# Acetate attenuates hyperoxaluria-induced kidney injury by inhibiting macrophage infiltration via the miR-493-3p/MIF axis

Wei Zhu[1,4], Chengjie Wu[1,2,4], Zhen Zhou[1,4], Guangyuan Zhang[3,4], Lianmin Luo[1], Yang Liu[1], Zhicong Huang[1], Guoyao Ai[1], Zhijian Zhao[1], Wen Zhong[1], Yongda Liu[1] & Guohua Zeng[1✉]

Hyperoxaluria is well known to cause renal injury and end-stage kidney disease. Previous studies suggested that acetate treatment may improve the renal function in hyperoxaluria rat model. However, its underlying mechanisms remain largely unknown. Using an ethylene glycol (EG)-induced hyperoxaluria rat model, we find the oral administration of 5% acetate reduced the elevated serum creatinine, urea, and protected against hyperoxaluria-induced renal injury and fibrosis with less infiltrated macrophages in the kidney. Treatment of acetate in renal tubular epithelial cells in vitro decrease the macrophages recruitment which might have reduced the oxalate-induced renal tubular cells injury. Mechanism dissection suggests that acetate enhanced acetylation of Histone H3 in renal tubular cells and promoted expression of *miR-493-3p* by increasing H3K9 and H3K27 acetylation at its promoter region. The *miR-493-3p* can suppress the expression of macrophage migration inhibitory factor (MIF), thus inhibiting the macrophages recruitment and reduced oxalate-induced renal tubular cells injury. Importantly, results from the in vivo rat model also demonstrate that the effects of acetate against renal injury were weakened after blocking the *miR-493-3p* by antagomir treatment. Together, these results suggest that acetate treatment ameliorates the hyperoxaluria-induced renal injury via inhibiting macrophages infiltration with change of the *miR-493-3p/MIF* signals. Acetate could be a new therapeutic approach for the treatment of oxalate nephropathy.

[1] Department of Urology and Guangdong Key Laboratory of Urology, The First Affiliated Hospital of Guangzhou Medical University, 510230 Guangzhou, Guangdong, China. [2] Breast Center, Department of General Surgery, Southern Medical University Nanfang Hospital, 510230 Guangzhou, Guangdong, China. [3] Department of Urology, Zhongda Hospital Southeast University, 210009 Nanjing, Jiangsu, China. [4] These authors contributed equally: Wei Zhu, Chengjie Wu, Zhen Zhou, Guangyuan Zhang. ✉email: gzgyzgh@vip.sina.com

Hyperoxaluria results from either inherited disorders of glyoxylate metabolism leading to hepatic oxalate over-production (primary hyperoxaluria) or increased intestinal oxalate absorption (secondary hyperoxaluria)[1]. Hyperoxaluria can cause not only nephrolithiasis and nephrocalcinosis, but also renal tubular damage, interstitial inflammation and fibrosis, and eventually end-stage renal disease[2,3]. Currently, available interventions aimed at the reduction of oxalate production include FDA-approved siRNA suppressing the expression of glycolate oxidase as well as pyridoxine in patients with primary hyperoxaluria[4,5] and the use of an oxalate-reduced diet and calcium supplementation in patients with enteric hyperoxaluria[6]. No therapies are yet known that blunt the effect of hyperoxaluria-induced inflammation and fibrosis in the kidney associated with renal failure.

Short-chain fatty acids (SCFAs) are end products from the fermentation of dietary fibers by the intestinal microbiota[7,8]. The most abundant SCFA is acetate. Recently, albeit limited, studies have attempted to use acetate therapeutically in animal and cell models of kidney injuries, such as ischemia-reperfusion-induced acute kidney injury, and diabetic nephropathy[9,10]. In our previous study, we first found exogenous acetate could improve renal function in rat models of hyperoxaluria[11,12]. This is independent on a decreased in urinary oxalate excretion and calcium oxalate crystals deposition in the kidney. However, how acetate ameliorates hyperoxaluria-induced renal injury and its underlying mechanisms remain incompletely understood.

Because hyperoxaluria nephropathy has an important inflammatory component yet acetate has anti-inflammatory properties, we investigated whether acetate treatment could protect rats from hyperoxaluria-induced kidney injury. Furthermore, we investigated whether this protection could involve direct modulation of the inflammatory process and or ameliorating of the macrophages infiltration in the hyperoxaluria rat model.

## Results

### Acetate-treatment ameliorates hyperoxaluria-induced renal injury and fibrosis.

We examined the effect of acetate on renal injury and fibrosis using an ethylene glycol (EG) induced hyperoxaluria. 8-week-old Sprague-Dawley rats received 1% EG in drinking water for 4 weeks to induce hyperoxaluria. In the meantime, rats were treated with 5% acetate (2 ml/kg) or distilled H₂O everyday by gavage. The results showed that acetate treatment diminished levels of serum creatinine and urea, and renal weight in hyperoxaluria rats while acetate treatment did not influence the urine oxalate levels (Fig. 1a, b). In addition, the increase in the percentage of necrotic tubules in the hyperoxaluria rats was significantly recovered after acetate treatment (Fig. 1c).

We further evaluated renal apoptosis, fibrosis, and inflammation by immunohistochemistry (IHC) and quantitative real-time PCR (Q-PCR). By IHC analysis, acetate treatment significantly reduced the level of renal tubular epithelial cell apoptosis and oxidative stress (Fig. 1c, TUNEL assay and 8-ohdg expression). The acetate treatment also decreased the number of myofibroblasts, the fibrotic area and interstitial collagen deposition in hyperoxaluria rats, which was shown by the anti-α-smooth muscle actin (αSMA) staining and Masson's trichrome staining (MTS) (Fig. 1c). The inflammatory cytokines (IL-1β and TNFα) were also diminished by acetate treatment. In addition, a low frequency of infiltrating macrophages (CD68⁺ and CD86⁺) was observed in acetate-treated rats (Fig. 1d). Q-PCR indicated a significant reduction in the expression of renal fibrosis-related genes (fn-1, *αSMA, Col1a1, Col3a1, Ctgf, fsp1, KIM*) and inflammatory cytokines (*Tgfα, Tgfβ, Il-1b, Il-4, Il-6, CD44, CD68*) in the kidney of the acetate-treatment group (Fig. 1e).

Results from Western blot also showed a reduction in the expression of renal fibrosis-related proteins (αSMA, Fn-1, Col3a1) and inflammatory factors (IL-1β, IL-6, TGF-β) in the kidney of acetate-treatment group (Fig. 1f).

In summary, results from Fig. 1a–f demonstrated that acetate treatment ameliorated the progression of renal injury, fibrosis, macrophages infiltration, and local inflammation in the hyperoxaluria rats.

### Acetate decreases oxalate-induced renal tubular cell injury via inhibiting macrophages infiltration.

As recent studies indicate that macrophage infiltration may play a key role in the progression of hyperoxaluria nephropathy, we were interested in testing the impact of macrophages on oxalate-induced renal cells injury[13]. We developed an in vitro system composed of renal tubular cells and macrophages to simulate hyperoxaluria conditions (see outline in Fig. 2a). Human renal epithelial HK-2 cells were co-cultured with phorbol 12-myristate 13-acetate (PMA)-induced THP-1 macrophages (MΦs). Oxalate was exposed to HK-2 cells after 48-h co-culture and the renal cell injury (cytotoxicity) was evaluated via measuring lactate dehydrogenase (LDH) activity. We found that HK-2 cells co-cultured with THP-1 cells in Transwell systems significantly increased the oxalate-induced renal cell injury (Fig. 2b). In addition, coculturing with THP-1 cells elicited remarkable upregulation of the proinflammatory cytokines *Ccl-2, Ccl-3, Ccl-4, Ccl-5* and *Il-1b* in HK-2 cells after exposure to oxalate condition (Fig. 2c). Similar results were observed when we replaced the HK-2 cells with mouse cortical collecting duct M-1 cells as well as replaced the THP-1 macrophages with mouse RAW264.7 macrophages (Fig. 2b, c). These data showed that macrophages could promote oxalate-induced renal inflammation and cells injury.

Next we tested the impact of the acetate on the macrophages infiltration to renal epithelial cells. We added acetate in HK-2 cells and examined the impact on the THP-1 macrophages recruitment to the renal epithelial HK-2 cells (see outline in Fig. 2d). Results showed that neither acetate nor oxalate affected the THP-1 macrophage recruitment in cell-free culture medium (Fig. 2e). When HK-2 cells exposure to oxalate, the THP-1 macrophage recruitment to the HK-2 conditioned medium (CM) increased significantly using the Transwell migration system. Conversely, adding acetate to the HK-2 cells inhibited the HK-2 cells CM capacity to more recruit the THP-1 macrophages. Similar results were also observed when we placed the THP-1 macrophages/HK-2 cells with RAW264.7 macrophages/M-1 cells (Fig. 2f).

Together, results from Fig. 2a–f demonstrated that acetate treatment in renal tubular cells inhibit the macrophages infiltration which could promote oxalate-induced renal inflammation and cells injury.

### Mechanism dissection of how acetate can alter the macrophages recruitment: via suppressing the macrophages migration inhibitory factor (MIF) expression.

To dissect the mechanism of how treatment of acetate in renal epithelial cells can inhibit the macrophages infiltration, we applied Western blot-based cytokine array analysis to screen inflammatory cytokines in HK-2 cells CM that potentially involved macrophages recruitment. The results revealed that the expression of MIF was altered most significantly after acetate treatment (Fig. 3a). In addition, we also confirmed the MIF protein expression of renal epithelial cells was also decreased after treatment of acetate via ELISA assay (Fig. 3b). Importantly, the MIF expression in the kidney was decreased after treatment of acetate in hyperoxaluria rats (Fig. 3c). We therefore decided to further study the impact of MIF on the acetate-altered macrophages recruitment.

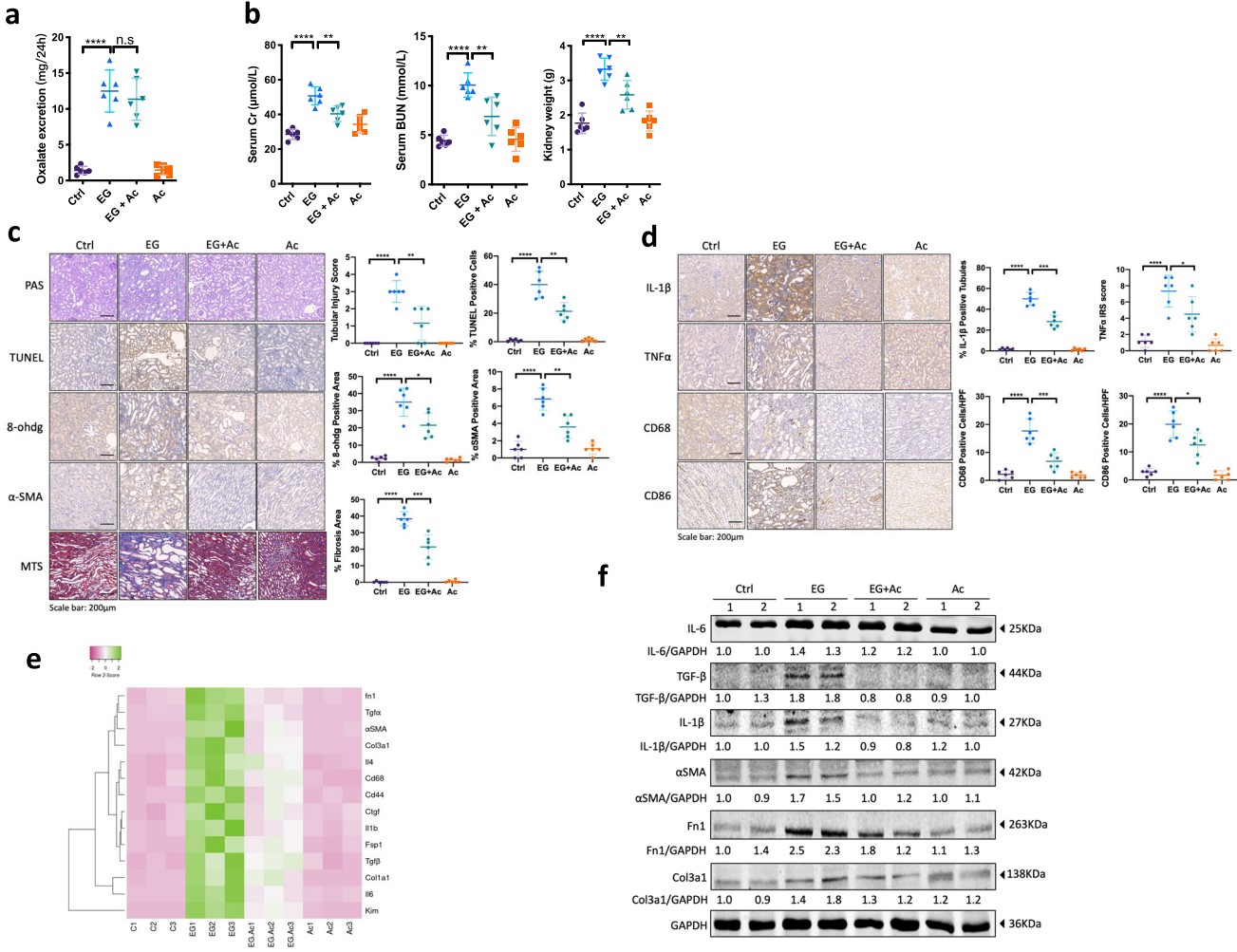

**Fig. 1 Acetate treatment protects against hyperoxaluria-induced renal injury. a** Detection of 24-h oxalate excretion in urine samples of each group of rats. **b** Serum creatinine and BUN levels, kidney weight, and renal damage degrade for each group. **c** Representative histologic kidney images of PAS, TUNEL, 8-ohdg, αSMA and Masson's trichrome stain (MTS). Renal damage was evaluated by scoring percentage of necrotic tubules in PAS sections. **d** Immunostaining of IL-1β, TNFα, CD68 and CD86 in kidney sections. **e** Relative transcript levels of genes in renal inflammation, fibrosis in the kidney tissue of each group of rats were measured using quantitative real-time PCR (q-PCR). **f** The expressions of renal inflammation factors and fibrosis-related protein in the kidney tissue were measured using Western Blot. $n = 6$ for each group. For (**c**) and (**d**), quantitations are at the right. Ctrl, control. EG, ethylene glycol. Ac, acetate. $*P < 0.05$, $**P < 0.01$, $***P < 0.001$, $****P < 0.0001$.

Using an interruption approach, we confirmed that treatment of recombinant MIF in renal epithelial cells could partially reverse the acetate-decreased macrophages recruitment, suggesting that treatment of acetate in renal epithelial cells may function via suppressing the MIF expression in the CM to inhibit MΦs recruitment (Fig. 3d). Similar results were also observed when we placed the THP-1 macrophages/HK-2 cells with RAW264.7 macrophages/M-1 cells (Fig. 3d).

Together, results from Fig. 3a–d suggest that treatment of acetate in renal epithelial cells may function via downregulating the MIF signals to inhibit the macrophages recruitment.

**Mechanism dissection of how acetate suppresses MIF protein expression: via upregulating the *miR-493-3p*.** The finding that acetate down-regulated MIF expression at the protein level but not at the mRNA level both in the animal model (Fig. 4a) and cultured cells (Fig. 4b) suggested that MIF expression is regulated at the post-transcriptional level, involving mechanisms such as differential miRNA expression. To directly test this hypothesis, we examined the expression of miRNAs that potentially regulated MIF based on the search of online

databases (DIANA-miRGen, MicroCosm Targets, RNA22) and published literature[12,14]. Results suggested that 5 miRNAs (*miR-363-5p*, *miR-493-3p*, *miR-629-3p*, *miR-1293*, *miR-1537-3p*) were likely candidates that were up-regulated by acetate in both in vivo (Fig. 4c) and in vitro models (Fig. 4d). We further assayed the consequences on MIF expression after directly transfection of these 5 miRNAs mimetics into HK-2 and M-1 cells, and results suggested that *miR-493-3p* was the best candidate for further study since altering this miRNA significantly suppress MIF expression (Fig. 4e, f).

As expected, results from the interruption approach via transient transfection with *miR-493-3p* antisense inhibitor led to partially reverse acetate-suppressed MIF expression in HK-2 and M-1 cells (Fig. 4g). The consequences of such reversion may then lead to partially reverse the acetate-decreased MΦs recruitment (Fig. 4h). In addition, transfection of miR-493-3p mimetic mimicked acetate effects in decreasing MΦs recruitment in HK-2 and M-1 cells (Fig. 4i).

Next, to directly test that the acetate-induced *miR-493-3p* could suppress the MIF expression, we searched and identified the potential binding sites of *miR-493-3p* in the 3′UTR of MIF

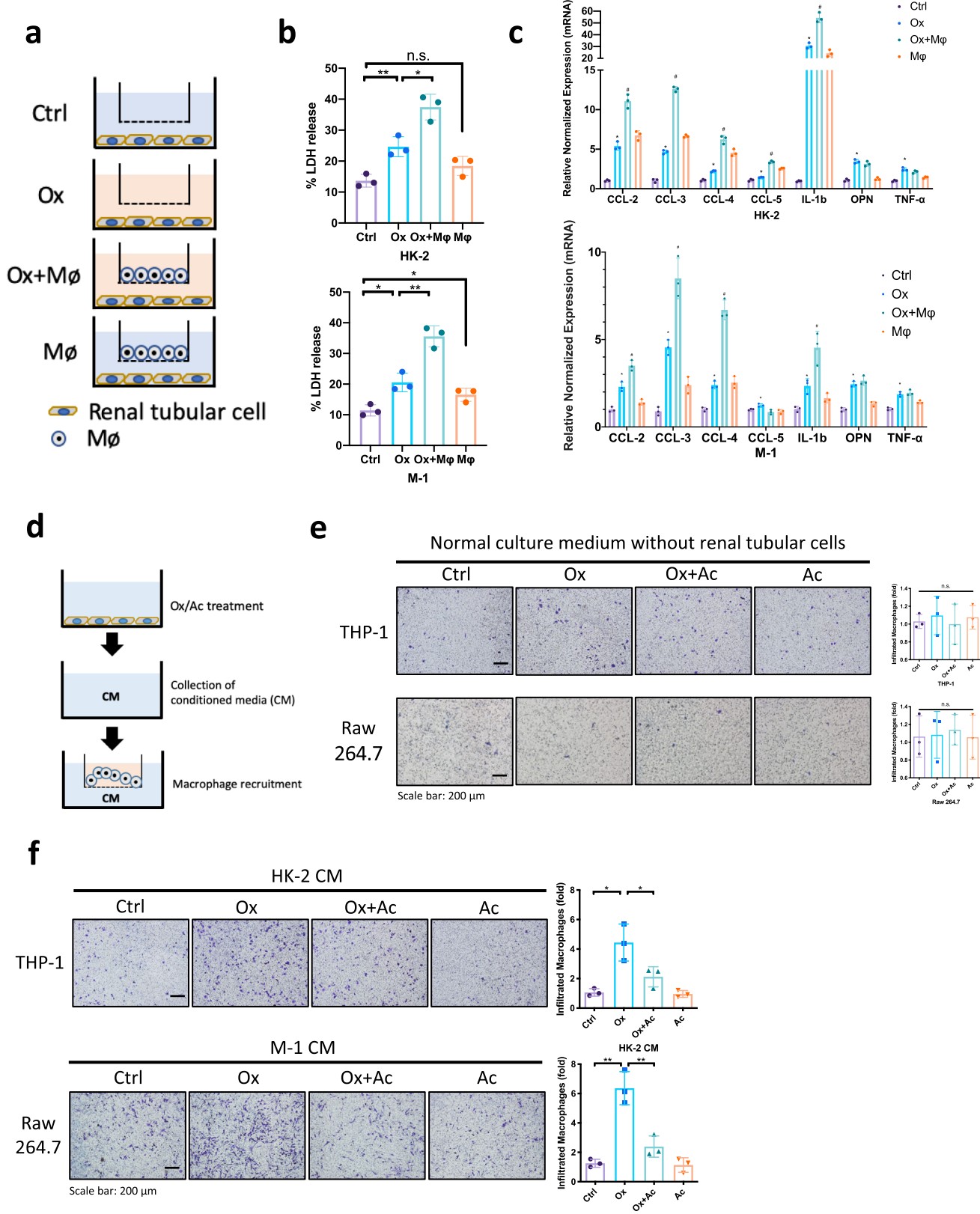

mRNA, and generated a luciferase reporter construct bearing the 3′UTR of MIF gene using a dual-luciferase reporter as well as a mutated version at the predicted target sites. The luciferase assay results revealed that *miR-493-3p* could suppress luciferase expression of the wild-type MIF 3′UTR construct, but not the mutant MIF 3′UTR construct, thus *miR-493-3p* could directly target MIF 3′UTR to suppress its expression (Fig. 4j).

Together, results from Fig. 4a–j suggested that acetate can suppress MIF protein expression via upregulating the *miR-493-3p* expression in the renal epithelial cells.

**Fig. 2 Acetate decreases oxalate-induced renal tubular cell injury via inhibiting macrophage infiltration. a** In co-culture system 4 groups were established, including Ctrl (control)—renal tubular cells cultivated in normal culture medium, Ox (Oxalate)—renal tubular cells cultivated in culture medium containing 0.5 mM oxalate, Ox+MΦ—renal tubular cells co-cultured with macrophages in culture medium containing 0.5 mM oxalate, MΦ—renal tubular cells co-cultured with macrophages in normal culture medium. **b** Lactate dehydrogenase (LDH) release measurement in renal tubular cells co-cultured with macrophages in 0.5 mM oxalate condition. **c** The Q-PCR analysis of inflammation-related gene expression in renal tubular cells co-cultured with macrophages in 0.5 mM oxalate condition. **d** Experimental outline for macrophages recruitment assay. The conditioned medium (CM) were collected from renal tubular cells treated with oxalate and/or acetate for 24 h. For macrophage recruitment assay, $1 \times 10^5$/well MΦs were added in the upper chambers, and the CM were placed into the lower chambers of transwell plates. **e** Macrophage recruitment to the normal culture medium without renal tubular cells. **f** The macrophages recruitment to CM from renal tubular cells treated with/without acetate were shown. $n = 3$ for each group *$P < 0.05$, **$P < 0.01$. #$P < 0.05$ compared with Ox group.

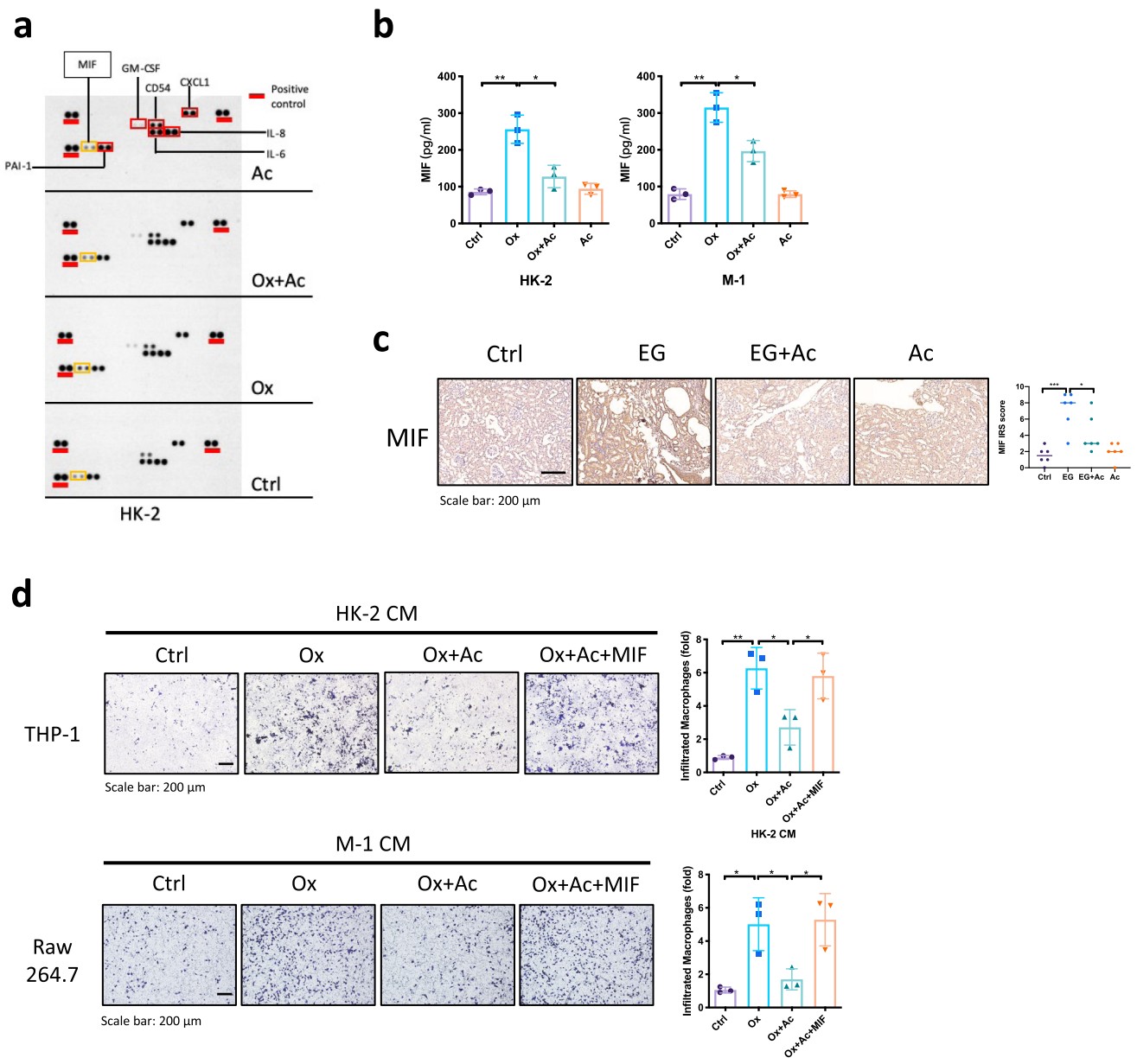

**Fig. 3 Acetate inhibits macrophage recruitment via modulating the MIF signals. a** Cytokine assay of different CM from HK-2 cells. CM of HK-2 treated with/without acetate were collected after 24 h incubation. MIF showed the most obvious decrease in CM from HK-2 treated with acetate (yellow squares). **b** The level of MIF in the CM of renal tubular cells was detected by ELISA. **c** Immunostaining of MIF in kidney sections of each group of rats. $n = 6$ for each group. **d** Treatment of recombinant MIF in renal epithelial cells could partially reverse the acetate-decreased MΦs recruitment. *$P < 0.05$, **$P < 0.01$.

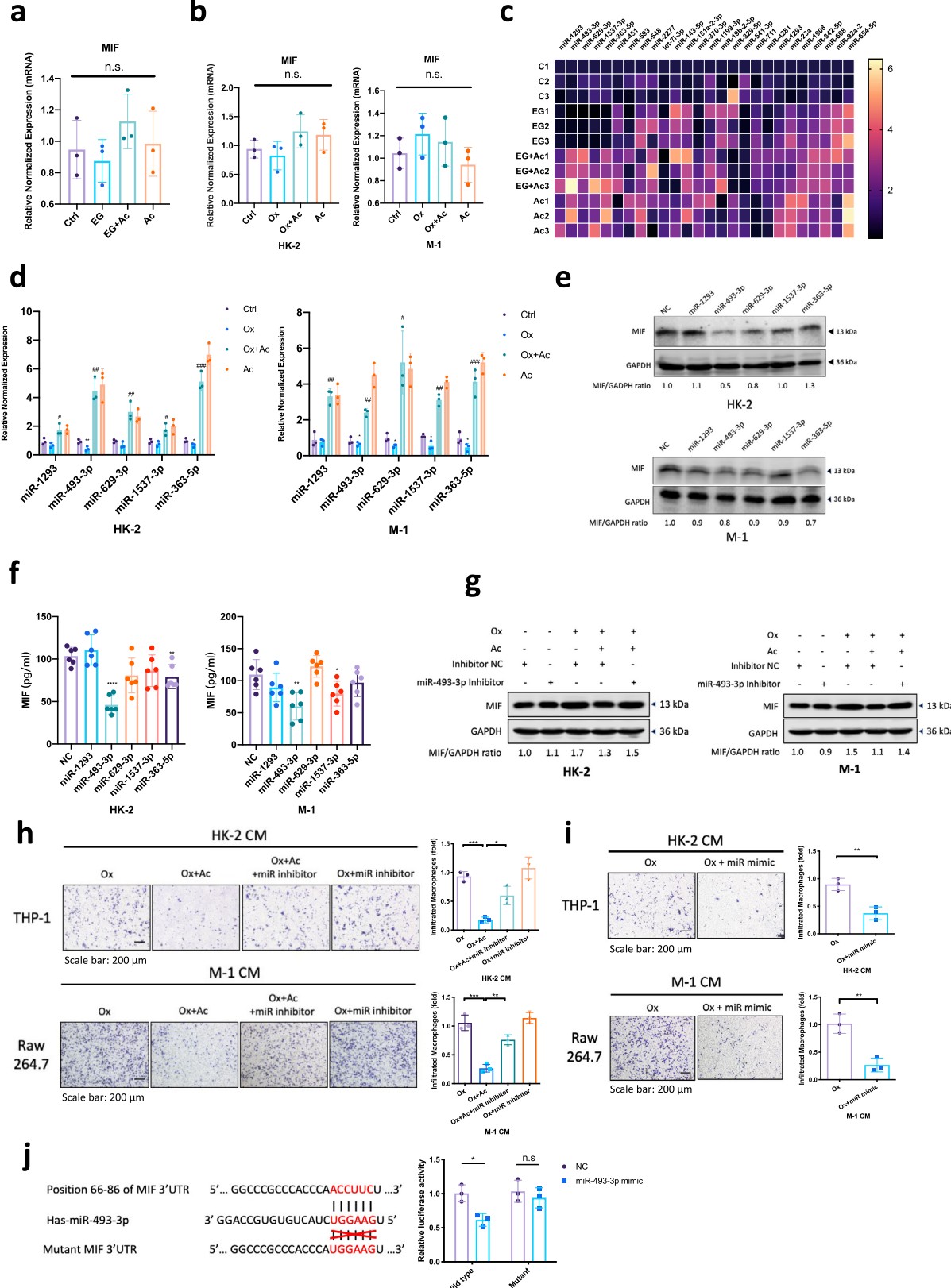

## Mechanism dissection of how acetate promotes miR-493-3p expression: via restoring histone acetylation.

To dissect the molecular mechanisms of how acetate promoted the expression of *miR-493-3p*, we first focused on histone acetylation since previous studies showed that acetate could function as an epigenetic metabolite to regulate gene expression[12,15]. We treated HK-2 and M-1 cells with acetate under hyper-oxalate conditions and found that acetate counteracted the decline of histone acetylation from hyper-oxalate treatment. Of particular interest, acetate induced a significant increase of H3K9 and H3K27 acetylation levels, but not H3K36 and H3K56 acetylation levels (Fig. 5a), indicating that acetate rescued oxalate-reduced histone acetylation with

**Fig. 4 Acetate modulates MIF via upregulation of *miR-493-3p* in renal tubular cells. a** q-PCR analysis of MIF mRNA expression in kidney from rats. **b** q-PCR analysis of MIF mRNA expression after 2 mM sodium acetate and/or 0.5 mM oxalate treatment for 24 h in HK-2 or M-1 cells. **c** 26 potential miRNAs candidates were screened by q-PCR assay in kidney from rats. **d** q-PCR analysis of 5 miRNA expressions after 2 mM sodium acetate and/or 0.5 mM oxalate treatment for 24 h in HK-2 or M-1 cells. **e** HK-2 or M-1 cells were transfected with 5 candidate miRNAs mimic or a negative control (NC). MIF expression was analyzed 48 h by Western blot. GAPDH serves as a loading control. **f** The protein expression levels of MIF in the CM of HK-2 and M-1 cells after transfection of 5 candidate miRNAs were assessed by ELISA. **g** HK-2 and M-1 cells were transfected with the miR-493-3p inhibitor or NC. 24 h later cells were treated with 0.5 mM oxalate and/or 2 mM sodium acetate. MIF expression was analyzed 24 h later by Western blot. **h** Macrophages recruitment to the CM from HK-2 cells (upper) and M-1 cells (lower) with four groups (0.5 mM oxalate, 0.5 mM oxalate + 2 mM sodium acetate, 0.5 mM oxalate + 2 mM sodium acetate + miR-493-3p inhibitor, 0.5 mM oxalate + miR-493-3p inhibitor). **i** Macrophages recruitment to the CM from HK-2 cells (upper) and M-1 cells (lower) with two groups (0.5 mM oxalate, 0.5 mM oxalate + miR-493-3p mimetic). **j** Co-transfection of MIF 3′UTR constructs containing wild type and mutant seed regions with miR-493-3p into HEK-293 cells and luciferase assay was applied to detect the luciferase activity. Ctrl, control. EG, ethylene glycol. Ac, acetate. Ox, oxalate. Data are from 6 rats in each group. n.s, not significant, *$P < 0.05$, **$P < 0.01$, ****$P < 0.0001$. #$P < 0.05$ compared with Ox group.

particular specificity. In addition, using IHC assays we found that EG treatment in rats significantly weakens renal H3K9ac and H3K27ac signal, and acetate treatment reversed this decrease (Fig. 5b).

To link the genome-wide histone acetylation change with locus-specific transcription of miR-493-3p, we carried out chromatin immunoprecipitation (ChIP)-qPCR assays for histones located at presumptive promoters of *miR-493-3p* and found that the acetylation levels (H3K9ac and H3K27ac) at *miR-493-3p* promoter were repressed after oxalate treatment, which were derepressed by acetate treatment in HK-2 cells (Fig. 5c).

Collectively, these data support the notion that acetate promotes the expression of *miR-493-3p* through epigenetic regulations.

**In vivo *miR-493-3p* is critical for the acetate effects in decreasing hyperoxaluria-induced kidney injury and fibrosis.** To directly test whether the acetate-induced *miR-493-3p* is mediating the effect of acetate in decreasing kidney injury and fibrosis in hyperoxaluria rat model, we performed an experiment in which chemically modified antisense oligonucleotides[12,16] specific to *miR-493-3p* (Antagomir-493-3p) was injected intraperitoneally (30 nmol/kg/week) into rats that had received 2 ml/kg/day 5% acetate by gavage and 1% EG in drinking water (see detail in Fig. 6a). The results reveled that treatment of antagomir-493-3p could partly reverse the acetate effects in diminishing levels of serum creatinine and urea, and renal weight in hyperoxaluria rats (Fig. 6b). As expected, antagomir-493-3p administration could also partly reverse the acetate effecting in improving renal apoptosis, fibrosis, and inflammation using IHC (Fig. 6c, d). In addition, IHC staining also showed that adding antagomir-493-3p led to reverse the effect of acetate-suppressed MIF expression (Fig. 6d). Finally, we assayed the infiltration of macrophages in renal tissues, and found acetate-treated rats had lower renal expression of CD68 and CD86 (two widely used marker of rat macrophages). As expected, adding antagomir-493-3p could then lead to partially reverse the acetate-suppressed macrophages infiltration (Fig. 6d).

These in vivo results from Fig. 6a–d confirmed that acetate diminished hyperoxaluria-induced kidney injury and fibrosis at least via upregulating the expression of *miR-493-3p*.

## Discussion

Hyperoxaluria can potential cause devastating consequences that can present as early as infancy or in the sixth decade of life and if not addressed appropriately, can cause significant morbidity and mortality including acute kidney injury and chronic kidney disease[2,3,17]. Macrophages-mediated inflammation plays an important role in regulating kidney injury and tissue fibrogenesis[13].

In the present study, we found acetate, which has antiinflammatory properties, could attenuates hyperoxaluria-induced kidney injury and fibrosis via inhibiting macrophages infiltrating (Fig. 7). To our knowledge, this is the first study demonstrating the protective role of acetate in oxalate nephropathy. Other studies have observed in a reduction in kidney injury in other models after acetate treatment[9,10,18].

Inflammation, apoptosis, and fibrosis are hallmarks of the hyperoxaluria rate model, and acetate treatment inhibited both processes. This is an expected result as acetate has previously been found to possess anti-inflammatory properties[10]. Infiltrating macrophages have long been known to be master players in inflammatory kidney diseases and to be associated with the development of kidney fibrosis and thereby failure of kidney function[19]. In our study, we found coculturing renal tubular cells with macrophages could increase the inflammatory cytokines levels and aggravate the oxalate-induced renal tubular cells injury. Increased CCL chemokines and IL-1β expression were detected in renal tubular cells coculturing with macrophages. IL-1β is strongly associated with the severity of tubulointerstitial lesions and renal impairment[20]. Some in vitro studies have demonstrated that IL-1β can exert potentially profibrotic effects such as stimulation of proliferation and extracellular matrix production[21].

The changes in inflammatory cytokines levels induced by coculturing renal tubular cells with macrophages without direct contact indicate that the crosstalk between macrophages and renal tubular cells is mediated by soluble factors released by these cells. In our study, we found MIF from renal tubular cells was elevated significantly after oxalate addition, and was suppressed by acetate treatment. MIF is an upstream proinflammatory cytokines and functions to initiate the inflammatory cascade response, and activates macrophages and T cells[22]. MIF was initially identified for its ability to inhibit the random migration of macrophages in vitro. However, recent evidence showed that MIF has pleiotropic effects on cell migration and chemotaxis[23,24]. Gregory et al. reported that MIF can induce macrophages recruitment through CCL2 and its receptor CCR2[25]. Hoi et al. found that renal macrophages recruitment and glomerular injury were significantly reduced in MIF knockout mice model, suggesting that MIF as a critical effector of organ injury in systemic lupus erythematosus[26].

Here, we first confirmed that MIF derived from renal tubular cells could play a key role in altering the renal macrophages and renal injury. We further proved that acetate could modulate MIF expression in renal epithelial cells, suggesting that altering the MIF expression via treating acetate to diminish the oxalate-induced renal injury is possible.

Acetate is an SCFA and has been reported to be readily absorbed in the intestines, transported into the blood stream, and

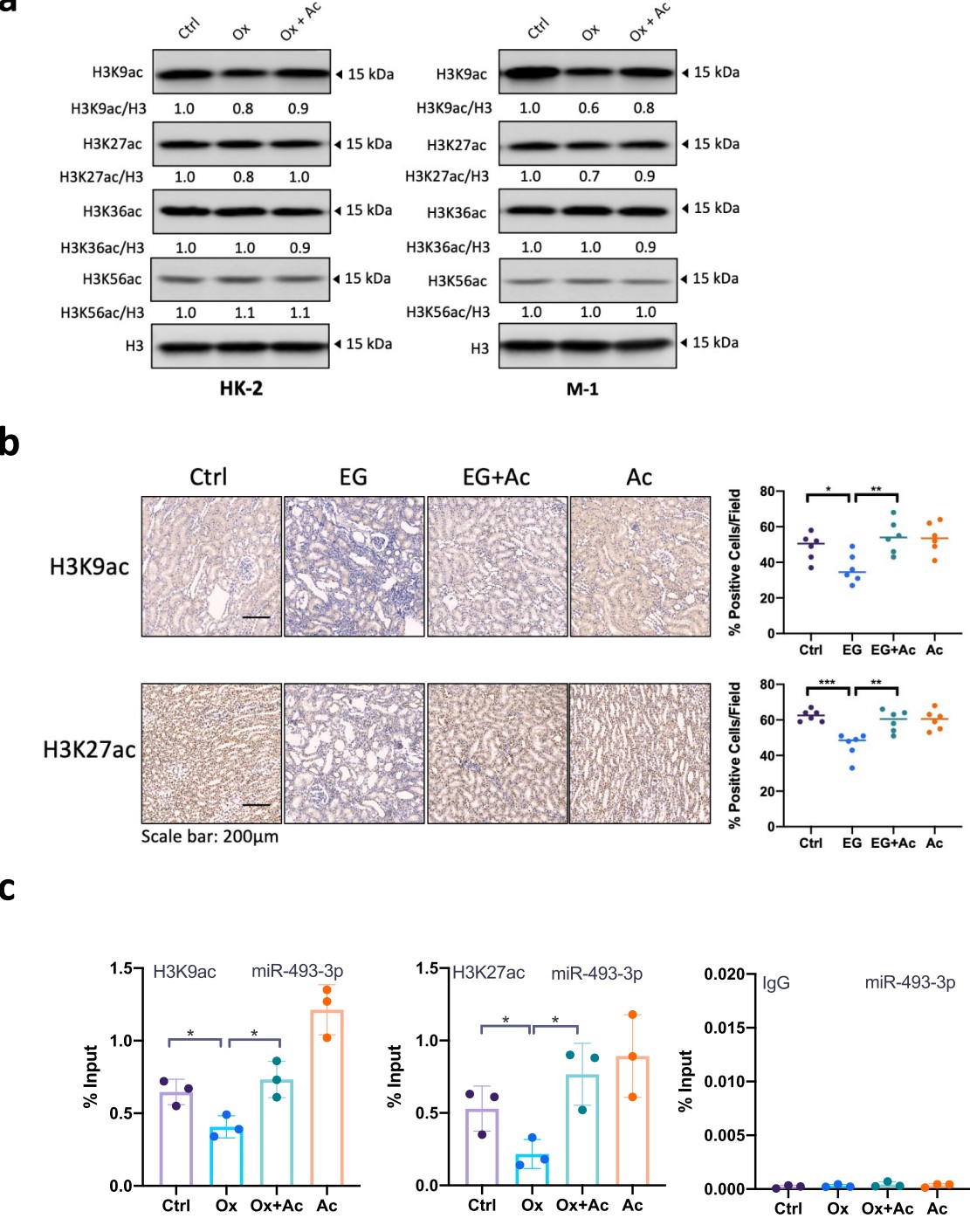

**Fig. 5 Acetate activated *miR-493-3p* through epigenetic regulation. a** Acetate rescued hyper-oxalate-reduced H3K9 and H3K27 acetylation levels. HK-2 or M-1 cells were treated with or without 2 mM sodium acetate under hyper-oxalate condition (0.5 mM) for 24 h. The histone acetylation levels were determined by Western blot. Total H3 served as a loading control. **b** IHC staining of H3K9ac and H3K27ac in kidney tissues from each group rats (amplification × 200). $n = 6$ for each group. **c** ChIP-qPCR assays showing histone acetylation enrich at miR-493-3p promoter region in HK-2 cells treated with or without 2 mM sodium acetate under normal or hyper-oxalate condition (0.5 mM) for 24 h. Rabbit IgG was included as negative control. For (**b**), quantitations are at the right. Ctrl, control. EG, ethylene glycol. Ac, acetate. Ox, oxalate. n.s, not significant, *$P < 0.05$, **$P < 0.01$,***$P < 0.001$.

easily incorporated in tissues[8,27]. Several studies reported that treatment of acetate could reduce kidney damage in different kidney injury animal models[10]. More evidence showed that acetate could through epigenetic regulation of histone acetylation in addition to its potential binding to G-protein membrane receptors (GPR41 and GPR43)[15,28]. Our previous study demonstrated that acetate could influence urinary compositions by regulating histone acetylation[12]. Constant with previous findings, in this study we found that acetate could influence macrophages infiltration to reduce oxalate-induced renal injury. It does so likely through regulating histone acetylation at H3K9 and H3K27 with a consequent activation of transcription of *miR-493-3p*, which in turn can suppress the expression of MIF, a key regulator of renal macrophages infiltration.

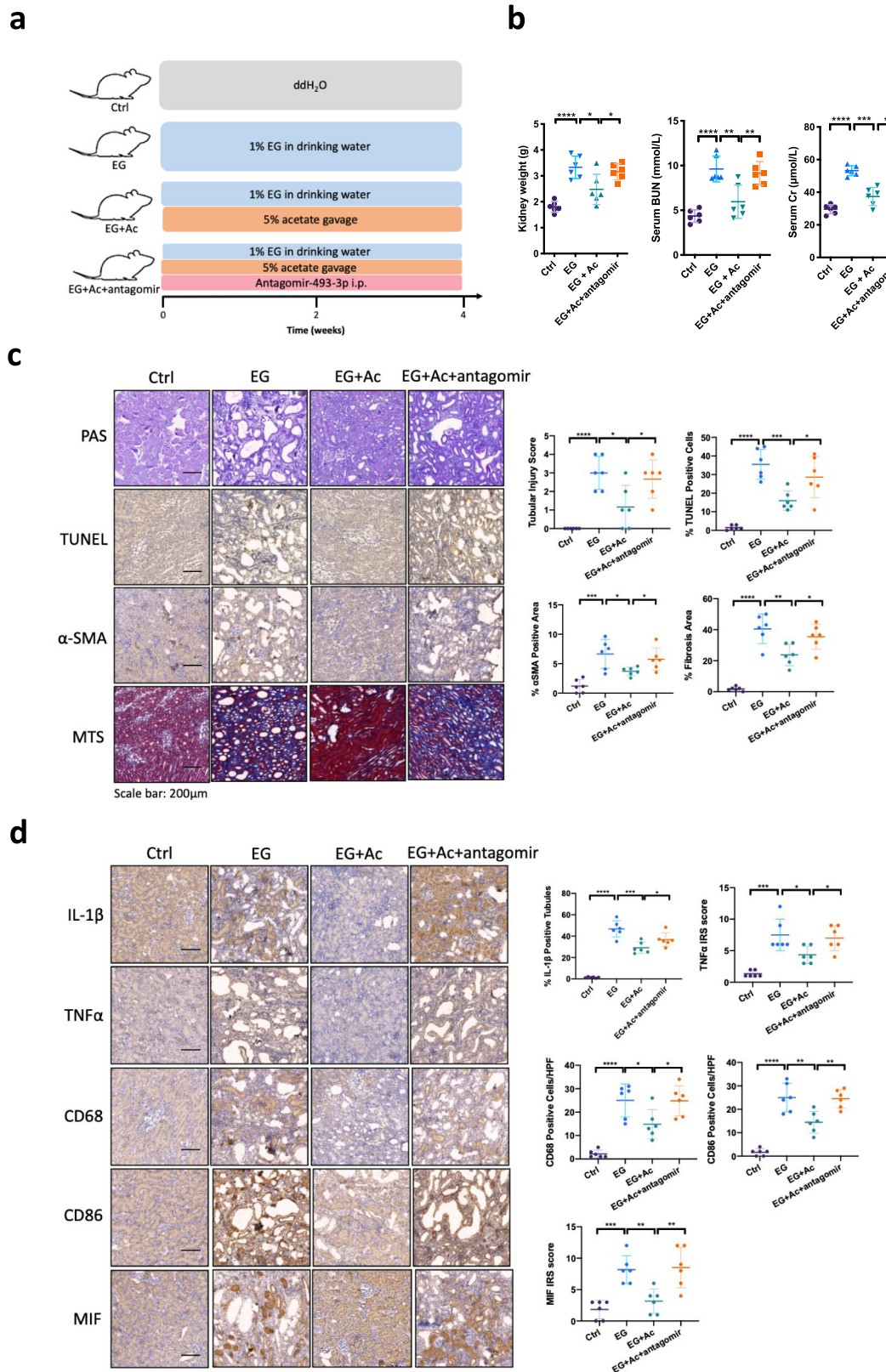

**Fig. 6 Antagomir-493-3p treatment attenuated acetate effects of regulating macrophages recruitment and decreasing renal injury. a** A diagram describing the injection schedule for antagomir-493-3p; EG, ethylene glycol; i.p., intraperitoneal injection. **b** Renal function was quantified by serum creatinine and BUN levels, and kidney weight. $n = 6$ for each group. **c** Representative histologic kidney images of PAS, TUNEL, αSMA and Masson's trichrome stain (MTS). **d** Immunostaining of IL-1β, TNFα, CD68, CD86, and MIF in kidney sections. $n = 6$ for each group. For (**c**) and (**d**), quantitations are at the right. *$P < 0.05$, **$P < 0.01$, ***$P < 0.001$, ****$P < 0.0001$.

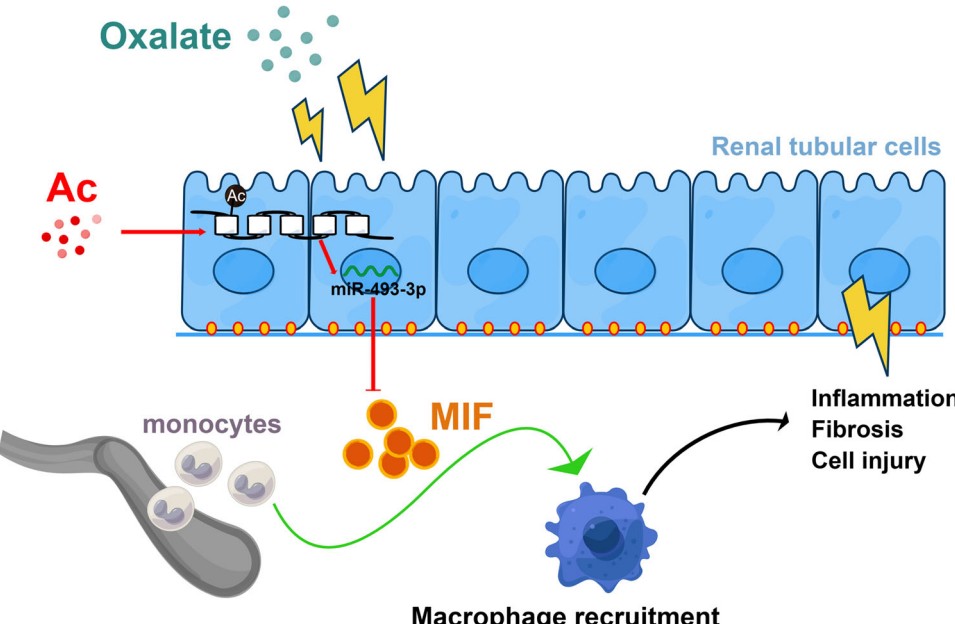

**Fig. 7 A scheme of acetate effect.** The scheme diagram summarizes the pathway described: acetate enhances the H3K9 and H3K27 acetylation levels at *miR-493-3p* promoter region, which downregulate MIF expression, and decrease macrophages infiltration to attenuate the hyperoxaluria-induced renal injury. The figure was created by Figdraw.

In summary, our study demonstrated that the treatment of acetate could function by altering the macrophages infiltration to influence oxalate-induced renal injury and fibrosis via altering the *miR-493-3p*/MIF signaling, which may provide clinicians a novel therapy to oxalate nephropathy.

## Methods

**Animal studies**. All rat experiments were performed under protocols approved by the Institutional Animal Care and Use Committee of the Guangzhou Medical University (Guangzhou, China).

**Development of hyperoxaluria rat model**. Male Sprague-Dawley rats, aged 6-8 weeks, were purchased from Guangdong Laboratory Animal Center. Rats were housed in polypropylene cages, and had access to food and water *ad libitum*. We established the hyperoxaluria rat model following the reported protocol[11,12,29]. Rats were given free access to food and drinking water containing 1% (v/v) EG for a period of 4 weeks. The rats were placed in metabolic cages for urine collection 1 day before sacrificing. Whole blood was collected and transferred to serum separator tubes and centrifuged to isolate the serum for further analysis. Serum BUN and creatinine were determined by Unicel DxC 600 synchronic biochemical detecting system. Urine oxalate was measured using ion exchange chromatography (Metrohm, Switzerland).

**Acetate treatment**. 8-week-old rats were divided into four groups. In the control group, animals were given tap water as their drinking water and 2 ml/kg ddH$_2$O by gavage for 4 weeks. The EG group animals were exposed to 1% EG in their drinking water and 2 ml/kg ddH$_2$O by gavage for 4 weeks. In the acetate group, rats were orally administered 5% (v/v) acetic acid dissolved with ddH$_2$O daily for 4 weeks.

**Antagomir treatment**. The miR-493-3p antagomir (5′-CUGGCACACAGUA-GACCUUCA-3′) and negative control antagomir were synthesized by Guangzhou RiboBio (Ribobio, China). Each antagomir was dissolved by autoclaved PBS according to the manufacturer's guidelines followed by ip injection to rats at dose of 30 nmol/kg body wt$^{-1}$.

**Histology and morphometric analyses**. Paraffin-embedded kidney pieces were cut into 5 µm sections and mounted on glass slides. The sections were depar-affinized with xylene, stained with MTS and periodic acid-Schiff (PAS). Tubular damage (epithelial necrosis) in PAS-stained sections was scored as follows: 0, normal; 1, <10%; 2, 10–25%; 3, 26–75%; 4, >75%. Tubular necrosis was defined as the loss of proximal tubular brush border blebbing of apical membranes, or intraluminal aggregation of cells and proteins[30]. At least 5 fields (magnification, ×200) were reviewed for each slide.

**Immunohistochemistry analysis (IHC)**. Kidney tissue were fixed in 10% for-maldehyde in PBS, embedded in paraffin, and cut into 5 *um* sections and used for histology and IHC staining with specific primary antibodies against 8-ohdg, α-SMA, IL-1β, TNFα, CD68, CD86, MIF, H3K9ac and H3K29ac. To enhance antigen exposure, the slides were treated with 10 mM sodium citrate (pH = 6 0) at 98 °C for 15 min for antigen retrieval. The slides were incubated with endogenous peroxidase blocking solution and then were incubated with the primary antibody at 4 °C overnight. After rinsing with PBS, the slides were incubated for 45 min with biotin-conjugated secondary antibody, washed, and then incubated with enzyme con-jugate horseradish peroxidase (HRP) streptavidin. Freshly prepared DAB (Zymed, South San Francisco, CA) was used as a substrate to detect HRP. Finally, slides were counterstained with hematoxylin and mounted with aqueous mounting media. The German immunoreactive score (IRS) (0 – 12) was calculated by mul-tiplying the percentage of immunoreactive kidney epithelial cells (0% = 0; 1–10% = 1, 11–50% = 2, 51–80% = 3; and 81–100% = 4) by staining intensity (negative = 0; weak = 1; moderate = 2; and strong = 3)[31]. The CD68, CD86, H3K9ac or H3K27ac- positive cells were determined using light microscopy. The antibodies used in this study are listed in Supplementary Table 1.

**TUNEL assay**. TUNEL was performed using the in situ Apoptosis Detection Kit (S7100-KIT; EMD milipore, CA, USA)[11]. Briefly, the paraffin-embedded sections were dewaxed. The sections were incubated in 0.3% H$_2$O$_2$ at room temperature to eliminate the endogenous peroxidase activity. Proteinase K was applied to the sections for 15 min at room temperature. TdT enzyme was applied to the sections and incubated in a humidified chamber for 1 h at 37 °C to allow extension of the nicked ends of the DNA fragments with digoxigenin-dUTP. Color was developed using 0.05% DAB with 0.006% H$_2$O$_2$ as substrate. For negative controls, distilled water was used instead of TdT enzyme.

**RNA extraction and quantitative real-time PCR (Q-PCR) analysis**. Total RNA was extracted by TRIzol reagent (Invitrogen) according to the manufacturer's instructions. RNAs (1 µg) were subjected to reverse transcription using Superscript III transcriptase (Invitrogen). Q-PCR was conducted using a Bio-Rad CFX96 system with SYBR Green to determine the mRNA expression level of a gene of interest. RNA expression levels were normalized to the expression of GAPDH. Primers used are in Supplementary Table 2.

For miRNA detection, 2 µg of total RNAs were subjected to reverse transcription using All-in-OneTM miRNA First-strand cDNA Synthesis Kit. Q-PCR was conducted using an All-in-OneTM miRNA qRT-PCR Detection Kits. Expression levels were normalized to the expression of 5 S rRNA or U6 snRNA.

**Cell lines and co-culture experiments**. The human proximal tubular epithelial HK-2 cells, human monocyte THP-1 cells, human embryonic kidney cell line HEK-293T, mouse macrophage RAW264.7 cells, and mouse cortical collecting duct M-1 cells were purchased from the American Type Culture Collection (ATCC) (Rockville, MD). The HK-2, RAW264.7, and M-1 cells were maintained in

Dulbecco's modified Eagle's media with 10% fetal bovine serum (FBS) and 1% penicillin/streptomycin. The THP-1 cells were cultured in RPMI-1640 media supplemented with 10% FBS. The THP-1 cells were differentiated to macrophages by treating with 100 ng/ml PMA for 3 days before being used in experiments. 6-well Transwell plates (3 μm) were used for co-culture experiments (Corning Inc., Corning, NY).

**Synthesis and transfection of miRNA mimics and inhibitors**. miRNA mimics and miRNA inhibitor were designed and synthesized by Guangzhou RiboBio (Ribobio, China). miRNA inhibitor was all nucleotides with 2'-O-methyl modification. 24 h prior to transection, cells were placed onto a 6-well plate at 40–60% confluence. Transfection was performed with riboFECTTM CP Reagent (Ribobio, China) according to the manufacturer's protocol. The medium was replaced 24 h after transfection with new culture medium.

**Renal tubular cells exposure to oxalate and/or acetate and collection of the CM**. Oxalate (Sigma) stock solution (10 mM) in PBS was diluted in medium to achieve final concentration of 0.5 mM. Sodium acetate (Sigma) stock solution (10 mM) in PBS was diluted in medium to achieve final concentration of 2 mM. HK-2 and M-1 cells were placed in 6-well culture dishes incubated overnight. The next day, the cell medium was changed to normal medium with 0.5 mM oxalate, and/or 2 mM sodium acetate. After 24 h of treatment, cells were collected for Q-PCR or western blot experiments, and the CM were collected for further experiments.

**Macrophages recruitment assay**. Chambers with 5.0 μm polycarbonate filter inserted in 24-well plates were used in the quantitative cell migration assays (Corning Inc., Corning, NY). In all, $1 \times 10^5$ PMA-differentiated-THP-1 macrophages or mouse RAW264.7 macrophages were plated onto the upper chambers, and the lower chambers were filled with the CM from HK-2 or M-1 cells. After 18- to 20-h incubation, the non-migrated cells in the upper chamber were removed and cells that migrated into the membrane were fixed with methanol, stained with crystal violet, and photographed under an inverted microscope. Cell numbers were counted in five randomly chosen microscopic fields per membrane. All experiments were performed in triplicate wells for each condition.

**Western blot**. Total protein was extracted by RIPA buffer containing 1% protease inhibitors (Amresco, Cochran, CA). Proteins (30-50 μg) were separated on 10% SDS/PAGE gel and then transferred onto PVDF membranes (Millipore). After blocking the membranes, they were incubated with appropriate dilutions (1:1000) of specific primary antibodies. The quantification was carried out by subtracting background from the band intensity of western blots by using Image J software.

**Human cytokine antibody array and ELISA**. CM was collected from HK-2. Relative amounts of cytokine levels were determined using Human Cytokine Array kit (ARY005B, R&D systems) according to the manufacturer's instructions. CM collected from culture cells were also used for detection of MIF by MIF ELISA kits (BOSTER) according to the manufacturer's instructions.

**ChIP-qPCR assay**. ChIP-qPCR assays were performed using a commercial kit (PierceTM Agrose ChIP Kit) according to the manufacturer's instructions. Briefly, $1 \times 10^7$ HK-2 cells were cross-linked with 1% paraformaldehyde, lysis and sonicated 13–15 times on ice until chromatin was 100–800 bps in size, with the center being ~300 bp. Solubilized chromatin was immunoprecipitated with ChIP grade antibodies for H3K9 acetyl, H3K27 acetyl or rabbit IgG (negative control). The DNA fragments were detected by qPCR. Histone acetylation marks were mapped at promoter spanning $-2$ to 2 kb of miR-493-3p. Primers spanning the regions with peaks were adopted for ChIP-qPCR analysis. Primers used are in Supplementary Table 2.

**Luciferase reporter assay**. Wild-type (WT) human MIF 3′UTR and mutated MIF 3′UTR (with a mutated sequence on the miR-493-3p binding site) were amplified from a human cDNA library. The 3′-UTR of MIF was constructed into psiCheck2 (Promega, Madison, WI, USA) by the Gibson assembly method. HEK293T cells were co-transfected with 25 ng/ml of either the luciferase reporter with WT or mutated 3′UTR, and 100 pmol of either miRNA mimics or miRNA negative control (NC). 48 h after co-transfection, a Dual-Luciferase Reporter Assay (Promega, USA) was carried out according to the manufacturer's protocol.

**Statistics and reproducibility**. All experiments were repeated independently, and statistical methods are described in the figure legends. p-values were determined by unpaired Student's t test using commercially available software (Prism 8) unless special methods were mentioned. $p < 0.05$ was considered statistically significant.

**Reporting summary**. Further information on research design is available in the Nature Portfolio Reporting Summary linked to this article.

## Data availability

Reasonable requests for additional data or materials will be fulfilled under appropriate agreements. All data generated or analyzed in this study are included in this published article. The source data underlying most graphs and charts used in this manuscript are provided as a Supplementary Data File. Uncropped and unedited blot images are provided as Supplementary Fig. 1. Request for any source data or materials that are not provided should be made to the corresponding author.

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

## Acknowledgements

This work was financed by grants from the National Natural Science Foundation of China (No. 8227031238 and No. 8187030283), the Young Talent Support Project of Guangzhou Association for Science and Technology, and the Guangzhou Science Technology and Innovation Commission (No. 202102010214). We thank Yin Sun and Xin Guo for helping in preparing the manuscript.

## Author contributions

Guohua Zeng has full access to all the data in the study and takes responsibility for the data and the accuracy of the data analysis. Study concept and design: Guohua Zeng and Wei Zhu. Acquisition of data: Wei Zhu, Chengjie Wu, Zhen Zhou, Guangyuan Zhang, Lianmin Luo, Zhicong Huang, Guoyao Ai, Yang Liu, Zhijian Zhao, Yongda Liu, Wen Zhong. Analysis and interpretation of data: Wei Zhu, Chengjie Wu, Zhen Zhou, Yang Liu. Drafting of the manuscript: Wei Zhu, Guohua Zeng. Critical revision of the manuscript for important intellectual content: Guohua Zeng. Statistical analysis: Wei Zhu, Chengjie Wu, Zhen Zhou. Obtaining funding: Guohua Zeng, Wei Zhu. Administrative, technical, or material support: Yongda Liu, Wen Zhong. Supervision: None. Other: None.

## Competing interests

The authors declare no competing interests.
