## [Peer Review File · Communications Biology]

Reviewers' comments:

Reviewer #1 (Remarks to the Author):

In this article, the authors found acetate could protect against hyperoxaluria-induced renal injury with less infiltrated macrophages. Moreover, the authors demonstrated acetate promoted expression of miR-493-3p by increasing H3K9 and H3K27 acetylation and the miR-493-3p suppressed the expression of MIF which inhibited the macrophages recruitment. Generally, this is a comprehensive study. However, there are still a few comments that the authors should address to strengthen the study.

Major comments:

1. In Figure 1, the authors should add Western-blot and ELISA experiments to detect fibrosis-related genes and inflammatory cytokines, not just IHC and PCR.
2. In Figure 2, the authors should add miRNA mimics group as another experimental group, especially miR-493-3p.

If the authors can take the above suggestions, I think this is a good study and can be published at any time.

Reviewer #2 (Remarks to the Author):

This is an interesting topic since until now treatment for oxalate-induced kidney injury is not available. I'm however obliged to mention here important shortcomings of this study/report.

1. I cannot find how many animals/wells are included in the different studies (number of animals/group, number of wells for each observation, are the in vitro experiments repeated?).
2. The complete research hypothesis/experimental set-up is based on the results shown in figure 1cd. There is however no quantification of any of the parameters shown in these figures, just 1 picture/treatment group is shown. Objective quantification is necessary (same for Fig 6cd)
3. With regard to the results shown in Figure 4gh: no quantification shown. To be honest I cannot observe the effects mentioned by the authors (partial reverse acetate-suppressed MIF expression-fig 4g and partially reversed acetate-decreased macrophages recruitment).
4. Idem for results shown in figure 5ab.
5. English grammar is not correct in the manuscript.

Answers to reviewers' critiques point by point

Reviewers' comments:

Reviewer #1 (Remarks to the Author):

Q1. In this article, the authors found acetate could protect against hyperoxaluria-induced renal injury with less infiltrated macrophages. Moreover, the authors demonstrated acetate promoted expression of miR-493-3p by increasing H3K9 and H3K27 acetylation and the miR-493-3p suppressed the expression of MIF which inhibited the macrophages recruitment. Generally, this is a comprehensive study.

Ans: Thanks for the positive comment.

Q2. In Figure 1, the authors should add Western-blot and ELISA experiments to detect fibrosis-related genes and inflammatory cytokines, not just IHC and PCR.

Ans: Thanks for the comments. As we are primarily concerned with the effect of kidney fibrosis in response to hyperoxaluria, we have added the experiment to detect the protein expression in addition to IHC and PCR through western blot of the kidney tissues, and not through ELISA as those factors will likely be diluted in circulation.

Q3. In Figure 2, the authors should add miRNA mimics group as another experimental group, especially miR-493-3p.

Ans: Thanks for the comment. As Figure 2 describes our effort to set up an in vitro system to dissect the potential mechanisms underlying the observations from Figure 1, the molecular details such as the identity of miRNAs that might play a role are yet to be determined, therefore it is not appropriate to add miRNA mimic in this setting. On the other hand, we now add miR-493-3p mimics as another experimental group in Figure 4 for the phenotype examined in Figure 2.

Reviewer #2 (Remarks to the Author):

Q1. I cannot find how many animals/wells are included in the different studies (number of animals/group, number of wells for each observation, are the in vitro experiments repeated?)

Ans: Thanks for the comment. We added the number of animals/wells in the revised manuscript. We confirmed that all experiments were performed in triplicate wells for each condition and repeated at least twice.

Q2. The complete research hypothesis/experimental set-up is based on the results shown in figure 1cd. There is however no quantification of any of the parameters shown in these figures, just 1 picture/treatment group is shown. Objective quantification is necessary (same for Fig 6cd)

Ans: Thanks for the comment. We added quantification in these figures accordingly.

Q3. With regard to the results shown in Figure 4gh: no quantification shown. To be honest I cannot observe the effects mentioned by the authors (partial reverse acetate-suppressed MIF expression-fig 4g and partially reversed acetate-decreased macrophages recruitment).

Ans: Thanks for the comment. We added quantification in these figures accordingly. From these data we can see that the miR-493-3p inhibitor could partly reversed the acetate effect.

Q4. Idem for results shown in figure 5ab.

Ans: Thanks for the comment. We added the quantification in this figure accordingly.

Q5. English grammar is not correct in the manuscript.

Ans: Thanks. We have revised whole manuscript carefully and tried to avoid any grammar of syntax error. We hope that the language is now acceptable for the next review procedure.

REVIEWERS' COMMENTS:

Reviewer #2 (Remarks to the Author):

The authors have made the necessary adaptations to their manuscript.

Final Revision Instructions

*To the Author— Please review the editorial comments and requests below and confirm that changes have been made in the manuscript in the right-hand column. **This document must be uploaded** as a related manuscript file.*

Please see our final file submission checklist for information about submitting your revised documents.

Files and General Policies	
Main manuscript file must be in Microsoft Word or LaTeX format. LaTeX and Tex article source files must be accompanied by the compiled PDF for reference. The bibliography must be submitted separately (as a .bib file) or contained within the .tex file.	Confirmed.
Each Figure must be provided as a separate file and must be supplied whole, with all panels included in a single document. Figures should be provided at a minimum resolution of 300 dpi at final size. Figure files must only contain images (please also leave out labels such as “Figure 1” etc). Figure captions must instead be included within the main manuscript file, grouped together at the end of the document.	Confirmed.
All figures, tables, and supplementary items must be cited in the manuscript and numbered in the order in which they appear.	Confirmed.
Please check whether your manuscript contains third-party images, such as figures from the literature, stock photos, clip art or commercial satellite and map data. We strongly discourage the use or adaptation of previously published images, but if this is unavoidable, please request the necessary	Confirmed.

rights documentation to re-use such material from the relevant copyright holders and return this to us when you submit your revised manuscript. An appropriate permissions statement must be present in the relative figure caption for any third-party images.	
Please check that you have not copied any text directly from published work (even your own) without clear attribution, including one or more references. We run a plagiarism detection software and may need to request additional changes if we identify large blocks of identical text.	Confirmed.
An updated editorial policy checklist that verifies compliance with all required editorial policies must be completed and uploaded with the revised manuscript. All points on the policy checklist must be addressed; if needed, please revise your manuscript in response to these points. https://www.nature.com/documents/nr-editorial-policy-checklist.pdf. Please note that this form is a dynamic ‘smart pdf’ and must therefore be downloaded and completed in Adobe Reader. This file will not open in an internet browser.	Confirmed.
The reporting summary will be published alongside your manuscript therefore it needs to accurately represent your work. In this case, please take a closer look at the reporting summary and make sure things are completed correctly. If an item does not apply, for example human participants, I need you to check the NA box next to that item. No section should be left blank. Also, please make sure to include your name and date at the top of the document. If you require a new Reporting Summary form, please download it here: https://www.nature.com/documents/nr-reporting-summary.pdf.	Confirmed.

Please note that this form is a dynamic 'smart pdf' and must therefore be downloaded and completed in Adobe Reader. This file will not open in an internet browser.	
Your paper will be accompanied by a brief editor's summary when it is published on our homepage. Please approve the draft summary below or provide us with a suitably edited version (no more than 250 characters including spaces). Acetate promotes miR-493-3p expression, which in turn suppresses the expression of macrophage migration inhibitory factor, leading to decreased macrophages recruitment and reduced hyperoxaluria-induced renal injury in mice.	Confirmed.
ORCID Communications Biology is committed to improving transparency in authorship. As part of our efforts in this direction, we are now requesting that all authors identified as 'corresponding author' create and link their Open Researcher and Contributor Identifier (ORCID) with their account on the Manuscript Tracking System (MTS) prior to acceptance. ORCID helps the scientific community achieve unambiguous attribution of all scholarly contributions. For more information please visit http://www.springernature.com/orcid. For all corresponding authors listed on the manuscript, please follow the instructions in the link below to link your ORCID to your account on our MTS before submitting the final version of the manuscript. If you do not yet have an ORCID you will be able to create one in minutes.	Confirmed.

https://www.springernature.com/gp/researchers/orcid/orcid-for-nature-research IMPORTANT: All authors identified as ‘corresponding author’ on the manuscript must follow these instructions. Non-corresponding authors do not have to link their ORCIDs but are encouraged to do so. Please note that it will not be possible to add/modify ORCIDs at proof. Thus, if they wish to have their ORCID added to the paper they must also follow the above procedure prior to acceptance. To support ORCID's aims, we only allow a single ORCID identifier to be attached to one account. If you have any issues attaching an ORCID identifier to your MTS account, please contact the Platform Support Helpdesk at http://platformsupport.nature.com/	
We regularly highlight papers published in Communications Biology on the journal’s Twitter account (@CommsBio). If you would like us to mention authors, institutions, or lab groups in these tweets, please provide the relevant twitter handles in the right-hand column.	No need.
We would welcome the submission of material for the ‘Featured Image’ section on the Communications Biology home page. Images should relate to the content of your manuscript but need not be contained within the paper. Photographs and aesthetically interesting images are preferred; diagrams are generally not used. Suggestions should be uploaded as a Related Manuscript file. Please provide 1200x675-pixel RGB images. You will also need to submit a completed Image License to Publish. Unfortunately, we cannot promise that your suggestions will be used.	Confirmed.
Supplementary information	

Supplementary Information Format and referencing  ● Supplementary Figures, small Tables, and any supplementary text must be provided in a single PDF. Figures and their captions should be presented together.  ○ If you include a title page, please check that the title and author list matches the main manuscript. ● All Supplementary items must be referred to in the manuscript, and items must be mentioned in numerical order. Please do not include general references to “Supplementary Material”; instead refer to specific items. ● Additional files can be provided as Supplementary Data (Excel files, text files, .zip folders), Supplementary Movies, Supplementary Audio, or Supplementary Software (.zip folder) Supplementary Information files will be uploaded with the published article as they are submitted with the final version of your manuscript. Any highlighting or tracked changes should be removed from the file.	Confirmed.
Supplementary items must be cited in a consistent format. Names of items in the Supplementary file(s) must match those used in the main manuscript. We recommend using the following naming formats: Supplementary Figure 1, Supplementary Table 1, Supplementary Data 1, Supplementary Note 1, and Supplementary References.	Confirmed.
It’s mandatory to provide access to the numerical source data (raw data) for all graphs and charts in the main and supplementary figures: We strongly recommend depositing these to suitable repositories (such as Figshare, Dryad, or a data type-specific repository if one exists).	

Otherwise, all source data underlying the graphs and charts presented in the main figures must be uploaded as Supplementary Data (in Excel or text format). Note that only the data used directly for generating the charts needs to be supplied. Please provide the source data as Supplementary Data 1. Please cite it in the Data Availability statement.	
For any Supplementary Files such as those mentioned above that are not included your combined PDF (e.g. Supplementary Data, Movies, Audio, Software), please provide a title and description for each file here in the column to the right. For example: File name: Supplementary Data 1 Description: The source data behind the graphs in the paper	
Title Page	
Please ensure that the author list provided in our manuscript tracking system matches the author list in the main manuscript.	Confirmed.
Manuscript title Please ensure the title clearly describes the central finding of the paper. We recommend writing the title as a declarative statement of approximately 15 words or fewer.	We revised the title according to the editor's suggestions.

Be sure to include any key species, protein names, or gene names to ensure optimal retrieval of the paper in database searches. The editors recommend the following title: Acetate attenuates hyperoxaluria-induced kidney injury by inhibiting macrophage infiltration via the miR-493-3p/MIF axis	
Abstract The abstract should be accessible to non-specialists and avoid jargon and abbreviations. Please write the abstract in the present tense. We recommend structuring the abstract as follows: Opening statement explaining why this area of research is important. A sentence explaining the gap in knowledge that your research will address. Here we show (or an equivalent phrase), and then the major results and conclusions of the paper. Final sentence indicating any broader impacts and how this research will be used in the future.	Confirmed.
Main text	
Format of the main text Please ensure your manuscript includes the following sections, presented in this order:  1. “Introduction”: The background and rationale for the work. The final paragraph should be a brief summary of the major results and conclusions. The results of the current study must only be discussed in 	Confirmed.

this final paragraph. The Introduction should contain no references to figures or tables. Do not include subheadings.  2. “Results” or “Results and Discussion”. This should be split into subheaded sections; we recommend 1 subheading per main figure or table. Figures should not be embedded in the text but submitted separately.  a. Do not use more than 1 layer of subheadings. b. A “Conclusions” paragraph can be included only if the results and discussion are combined into a single section. 3. “Discussion” (optional), without subheadings. 4. Methods, which should be split into subheaded sections. Do not use more than 1 layer of subheadings. To improve readability, we recommend that the main text (Introduction, Results and Discussion) be limited to approximately 5000 words or fewer. Please retitle the Methods section.	
Statistical reporting Wherever statistics have been derived (e.g. error bars, box plots, statistical significance) the legend needs to provide and define the n number (i.e. the sample size used to derive statistics) as a precise value (not a range), using the wording “n=X biologically independent samples/animals/independent experiments” etc. as applicable. Please provide and define the n number for all relevant figures.	Confirmed.
Statistical representation	Confirmed.

Statistics such as error bars cannot be derived from $n < 3$ and must be removed from all such cases. We strongly discourage deriving statistics from technical replicates, and they should be removed from all such cases, unless there is a clear scientific justification for why providing this information is important. Conflating technical and biological variability, e.g. by pooling technically replicate samples across independent experiments is strongly discouraged.	
Avoid the use of the word “significant” unless referring the results of a statistical test. In Line 92, please either remove “significant” or provide statistical analysis with $n \geq 3$.	Confirmed.
Please check that all gene and mRNA names are in italics. Protein names should not be in italics. Please confirm that only official gene/protein symbols are used and that species names are in italics.	Confirmed.
Language such as “new”, “novel”, etc, should be avoided, or qualified with “to the best of our knowledge” or similar, because it often leads to unproductive controversy. Novelty should be made clear from the context.	Confirmed.
We recommend editing the main text for English language and grammar to improve readability and clarity for our readers. If you would like the assistance of paid editing services to do this, we can recommend our affiliates, Nature Research Editing Service: https://authorservices.springernature.com/language-editing and American Journal Experts: https://www.aje.com/go/springernature	Confirmed.

Please note that use of an editing service is neither a requirement nor a guarantee of publication. Free assistance is available from our resources page: https://www.springernature.com/gp/researchers/campaigns/english-language-forauthors	
Display items	
Figure captions/legends Figures must have a title that will appear above the Figure and a legend that will appear below the Figure (see e.g. https://www.nature.com/articles/s42003-020-1059-1/figures/1) The Figure title must describe the Figure as a whole and must not contain reference to specific figure panels. The Figure legend must refer to and describe all panels. Abbreviations, symbols, colors, and shading present in the Figure must be defined. Please write out the symbols/colors in words (blue circles, red dashed line, etc.) within these definitions. All figure panels must be labelled using lower case letters. Please refrain from referring to sections of figures as top/bottom/left/right/, etc. Please enlarge the panels in Fig. 1c, 1d, 5b, 6c, 6d on the right.	Confirmed.
Axis and panel labels will be published as received. We recommend using a sans-serif font such as Arial or Helvetica.	Confirmed.
Data presentation in bar graphs and line graphs	Confirmed.

For all graphs depicting a single point value (e.g., mean) with error bars, **you must add individual data points or convert the graph to a boxplot or dot-plot**. You may wish to refer to this blog post about representing data distribution in plots (particularly for small datasets). We strongly encourage the same for plots with multiple time courses depicted. See the June 24, 2019 CommsBio editorial for more details about this policy. Example plots are shown here:

Examples of plots showing data distribution. Figure 2 from the editorial linked to above.

Please provide individual data points for all non-compliant graphs.

Microscopy images and photographs in each Figure and Supplementary Figure must be accompanied by scale bars, and these must be defined. Please add scale bars and corresponding definitions to Fig. 2, 3d, 4, 5b, 6c, 6d.	Confirmed.
Blots and gels All blots/gels must be accompanied by size markers in every figure panel. Uncropped and unedited blot/gel images must be included as Supplementary Figure(s). The new Supplementary Figure(s) must be cited in the main manuscript text (for example, in the Data Availability Statement). Please pay close attention to our Digital Image Integrity Guidelines and to the following points below:  ● that unprocessed scans are clearly labelled and match the gels and western blots presented in figures. Unprocessed scans must be included in a supplementary figure. ● that control panels for gels and western blots are appropriately described as loading on sample processing controls ● all images in the paper are checked for duplication of panels and for splicing of gel lanes. Finally, please ensure that you retain unprocessed data and metadata files after publication, ideally archiving data in perpetuity, as these may be requested during the peer review and production process or after publication if any issues arise. Please include size markers for all western blots. Please provide uncropped and unedited blot images as Supplementary Figure 1 and please cite it in the Data Availability statement.	Confirmed.

Methods	
Please ensure that all information present in the Reporting Summary is also in the manuscript. This information is usually most appropriate in the Methods section.	Confirmed.
We allow unlimited space for Methods. The Methods must contain sufficient detail such that the work could be repeated. It is preferable that all key methods be included in the main manuscript, rather than in the Supplementary Information. Please avoid use of “as described previously” or similar in Line 335, and instead detail the specific methods used with appropriate attribution.	Confirmed.
The Methods should include a separate section titled “Statistics and Reproducibility” with general information on how the statistical analyses of the data were conducted, and general information on the reproducibility of experiments (also those lacking statistical analysis), including the sample sizes and number of replicates and how replicates were defined. Please provide general info on the reproducibility.	Confirmed.
For studies using live vertebrates , a statement affirming that you have complied with all relevant ethical regulations for animal testing and research is necessary. A statement explicitly confirming if the study received ethical approval, including the name of the board and institution that approved the study protocol is also required. The species, strain, sex and age of animals should be included.	Confirmed.
Data Policies	

The Data Availability statement must include:  ● Access details for deposited data, including repository name and unique data ID. ● How source data can be obtained. ● A statement that all other data are available from the corresponding author (or other sources, as applicable) on reasonable request. Note that ‘available upon request’ is only appropriate if immediate data access has not been mandated by our policies or by the editors. See here for more information about formatting your Data Availability Statement: http://www.springernature.com/gp/authors/research-data-policy/data-availability-statements/12330880	Confirmed.
Communications Biology has a strong preference for all data to be deposited in an approved repository. In some cases, data deposition may be required by the editor. We recommend the following data repositories:  ● GenBank (all DNA sequence data) ● NHGRI-EBI GWAS Catalog (GWAS summary statistics) ● PGS Catalog (polygenic risk scores) ● Gene Expression Omnibus (Microarray or RNA sequencing data) ● Sequence Read Archive (WGS or WES data) ● Protein Data Bank (protein structural data) ● OSF (neuroimaging raw data and EEG/EMG/MEG raw data) ● Neurovault (unthresholded statistical maps, parcellations, and atlases produced by MRI and PET studies) ● Image Data Resource (microscopy data) ● PRIDE (proteomics data) 	Confirmed.

Data types without a specific repository can be deposited in a generalist repository, such as figshare or Dryad. For an up-to-date list of approved repositories, please visit https://www.springernature.com/gp/authors/research-data-policy/repositories/12327124.	
Data citation Please cite datasets stored in external repositories in the main reference list. For previously published datasets, we ask authors to cite both the related research articles and the datasets themselves. For more information on how to cite datasets in submitted manuscripts, please see our data availability statements and data citations policy.	Confirmed.
End Notes	
Please check that your bibliography complies with the following:  ● Your bibliography should start with the heading “References”. The references must be numbered in the order of appearance in the text, then tables, then figures. ● Any in-text citations to references (e.g. "Gupta et al. show...") should be followed by their corresponding reference citation number from the reference list. ● Manuscript citations must include journal title, article title, volume number, page or article number or DOI, and year of publication. ● No publication can be present more than once in the reference list. 	Confirmed.

 • No footnotes are permitted in the references or elsewhere. Text should be incorporated into the main text, the Methods section, or the Supplementary Information instead. • Websites should only be listed in the references if they are in common use or curated. • Where possible, preprints in the reference list should be updated with details of the published, peer-reviewed paper. • Citations should be formatted in the text using superscript numbers. 	
Please provide a 'Competing interests' statement using one of the following standard sentences:  • The authors declare the following competing interests: [specify competing interests] • The authors declare no competing interests. See our competing interests policy for further information: https://www.nature.com/nature-research/editorial-policies/competing-interests Please use the standard title and sentence.	Confirmed.
Please check that your 'Author Contributions' section individually lists the specific contribution of each author to the work. Each author must be referred to by name or initials. Where multiple authors possess identical initials, they must be clearly disambiguated from one another. See our author contributions policy for further information: https://www.nature.com/nature-research/editorial-policies/authorship#author-contribution-statements	Confirmed.

communications
biology